# Interactions between Blackcurrant Polyphenols and Food Macronutrients in Model Systems: In Vitro Digestion Studies

**DOI:** 10.3390/foods10040847

**Published:** 2021-04-13

**Authors:** Elena Diez-Sánchez, Amparo Quiles, Isabel Hernando

**Affiliations:** Department of Food Technology, Universitat Politècnica de València, Camino de Vera s/n, 46022 València, Spain; eldiesan@upvnet.upv.es (E.D.-S.); mquichu@tal.upv.es (A.Q.)

**Keywords:** bioaccessibility, total phenolic content, microstructure, dietary fiber, antioxidant capacity

## Abstract

Blackcurrant pomace, rich in fiber and polyphenols, can be used as added-value ingredient for food formulation. However, the bounding of polyphenols to pomace and the interactions that take place with food nutrients modify polyphenol bioaccessibility. This work studied the interactions between polyphenols and the main macronutrients in foods, and the changes that occurred during in vitro digestion, using model systems. Model systems were formulated with (i) water, (ii) wheat starch, (iii) olive oil, (iv) whey protein, and (v) a model combining all the ingredients. Polyphenols were added from two sources: as pomace and as a polyphenolic pomace extract. Interactions between polyphenols and macronutrients were studied using light microscopy; total phenolic content (TPC) and antioxidant capacity (AC) were determined before and after the in vitro digestion process. Lastly, the bioaccessibility of the samples was calculated. Polyphenols incorporated into the model systems as pomace increased their bioaccessibility if compared to polyphenols added as extract. For single-nutrient model systems formulated with pomace, the bioaccessibility was higher than when the system contained all the nutrients. Of all the components studied, the greatest effect on bioaccessibility was observed for proteins.

## 1. Introduction

By-products of the industries of fruit juice processing, also known as pomaces, comprise the remains of skins and seeds and are rich in bioactive compounds. These bioactive compounds include dietary fiber, polyphenols, carotenoids, and other phytochemicals [1].

Polyphenolic compounds are characterized by the presence of one or more hydroxyl groups linked to aromatic rings and can vary in their chemical structure and properties, ranging from simple molecules, such as phenolic acids, to highly polymerized molecules, such as proanthocyanidins; this structure gives them their antioxidant capacity (AC). Several studies relate the AC of polyphenols with prevention of certain types of cancer [2,3] and other diseases, such as type 2 diabetes [4]. This protective activity has led to the study of the benefits of its consumption for health [5,6,7,8].

Berry pomace is a valuable source of dietary fiber with associated phenolic compounds, which could be a good alternative for its use as a value-added ingredient in functional foods. The main anthocyanins detected in pomaces from blackcurrant are cyanidin 3-galactoside, cyanidin 3-glucoside, cyanidin 3-arabinoside, and cyanidin 3-xyloside [9]. The further use of pomace in food formulations contributes to improve the sustainability of the agri-food processing chain. Moreover, their great variety, effect on health, and relative ease to obtain make berry pomace polyphenols the ideal bioactive compounds to produce functional foods. However, polyphenols in pomaces are linked to the fiber matrix. This association can be conducted by hydrogen bonds formed between polyphenol hydroxyl groups and the fiber hydroxyl groups, or between polyphenol hydroxyl groups and the fiber oxygen atom of the glycosidic linkages of polysaccharides. Van der Waals forces are also possible because both polyphenols and dietary fibers possess polarizable molecules. Furthermore, hydrophobic interactions can take place between the hydrophobic sites of polysaccharides and the aromatic rings of polyphenols. Finally, they can be linked by covalent bonds, such as ester bonds, between the phenolic acids and polysaccharides [10,11].

When these value-added ingredients are incorporated into food matrices, it is necessary to consider that the hydrophobic aromatic ring and the hydrophilic hydroxyl groups of the polyphenols interact with different macromolecules from food ingredients. These interactions cause reversible interactions through non-covalent forces, such as Van der Waal forces, hydrogen bonding, and hydrophobic interactions, and irreversible interactions through covalent bonding [12,13,14,15,16]. The polyphenol reactivity toward these forces will depend on structural factors, including the molecular weight, conformational mobility, flexibility, projection of the hydroxyl groups, the presence of galloyl groups, or the polyphenol’s affinity for water [15].

To optimize the food formulation, it is important to study the digestibility of its components. The efficacy in the absorption through the gastrointestinal tract of the polyphenols is affected by several factors, such as the food matrix, solubility, digestibility, bioaccessibility, molecular structures, or metabolizing enzymes [17,18,19]. Moreover, adding the polyphenols as a pomace or extract also influences their bioaccessibility. Although in vivo digestions best represent the digestion of food in real conditions, in vitro digestions are a cheaper, reproducible, and faster alternative for obtaining first hypotheses regarding the digestion of a certain food or nutrient [20,21]. Some in vitro studies describe interactions between polyphenols and the different macronutrients from foods [12,14], such as carbohydrates [22,23], lipids [24], and proteins [25,26,27].

In the simulated in vitro digestion, the process is divided into three key steps, which are the oral phase (mouth), gastric phase (stomach), and small intestine (jejunum). In the oral phase, the macrostructure breakdown and bolus formation take place due to the mechanical action and the enzyme activity within the food matrix. The latter is limited to food rich in carbohydrates, in which starch is minimally hydrolyzed by the α-amylase enzyme [28,29,30]. This phase is where the bioconversion begins [31]—the glycosidase starts to hydrolysate the glycosidic flavonoids—but its effectiveness depends on the sugars present in the molecule; e.g., glucose conjugates are rapidly hydrolyzed in contrast to rhamnose conjugates [19]. The gastric phase occurs in an acidic environment with the presence of the pepsin enzyme. Due to low pH conditions, the anthocyanins transform to a stable species, which is the optimum for preserving the anthocyanins’ natural structure, as flavium cations, avoiding its degradation [17,31]. In these conditions, the polymerized polyphenols with a high molecular weight (insoluble) are hydrolyzed to monomers or aglycones (soluble) due to pH changes [32]. In the small intestine, the polyphenols are sensitive to the mild alkaline conditions; thus, their stability decreases, being transformed into different structural forms with different chemical properties. Therefore, after the intestinal digestion, new anthocyanin degradation products (such as phenolic acids) and other forms of anthocyanins (such as chalcones) are generated [33]. Finally, the insoluble fraction would reach the colon where they are absorbed unaltered (through the epithelium) or metabolized by colonic microbiota [34].

When functional foods are prepared using value-added ingredients, such as polyphenols, their interaction with the different food macronutrients must be considered. Moreover, these interactions can be modified during the digestion process by the conditions and reactions that occur throughout it. Thus, the objective of this work was to study the interactions between polyphenols and macronutrients in different model systems and the implications of these interactions in the bioaccessibility of the polyphenols. For this purpose, two sources of polyphenols were used: dried berry pomace, where the polyphenols are bounded to fiber, and a polyphenolic extract obtained from blackcurrant pomace.

## 2. Materials and Methods

### 2.1. Materials

The ingredients used to prepare the model systems were dried blackcurrant (*Ribes nigrum*) pomace (BCP), supplied by the Institute of Natural Materials Technology (Technische Universität Dresden, Germany) and prepared by drying the fresh pomace at 70 °C for 2 h and milling it in a ZM 100 ultracentrifuge mill (Retsch GmbH, Haan, Germany) at 14,000 rpm using a 1 mm sieve [9] (Reißner et al., 2019). The blackcurrant extract (BCE) was prepared from BCP, olive oil (Consum, Valencia, Spain), native wheat starch (C*Gel, Cargill BV, Amsterdam, Netherlands), and whey protein isolate from milk (Harrison Sport Nutrition S.L, Granada, Spain). All the enzymes (α-amylase from porcine pancreas, pepsin from porcine gastric mucosa, porcine bile extract, lipase from porcine pancreas, and pancreatin from porcine pancreas) used in the in vitro digestion analysis were supplied by Sigma-Aldrich (Barcelona, Spain).

### 2.2. Polyphenol Extraction from Blackcurrant Pomace

To obtain the polyphenolic extract from dried BCP, first an extraction was conducted for 120 min at 60 °C in 60% ethanol, in agitation and darkness with a 1:8 sample-to-solvent ratio [35]. After the extraction, vacuum filtering was conducted, and the ethanol was removed with rotary evaporation. The resulting aqueous extract was freeze-dried (Telstar, Azbil group, Terrassa, Spain) to obtain a polyphenol-rich extract powder (BCE).

### 2.3. Model Systems Preparation

The consumption data in Spain of the different macronutrient are 5.5 g of carbohydrates, 4 g of fat, and 4 g of protein per 100 g of consumption [36]. According to this information, five types of model system were prepared with (i) water, (ii) wheat starch as a standard for carbohydrates, (iii) olive oil as a standard for fats, (iv) whey protein as a standard for proteins, and (v) a model combining all the ingredients described (water, wheat starch, olive oil, and whey protein). Five grams of dried pomace (BCP) or 0.275 g of polyphenol-rich extract (BCE) (corresponding to the equivalent amount of polyphenols in 5 g of pomace) were added to the different model systems; ten model systems were studied.

The models were prepared as follows. The water model system was prepared by dissolving 5 g of BCP or 0.275 g of BCE in 100 mL of distilled water for 30 min and the obtained systems were called BCP-Wa and BCE-Wa, respectively. For the carbohydrate model, 5.5 g of starch were gelatinized in distilled water (up to 100 mL) for 40 min at 65 °C, with constant agitation, and after cooling the sample, BCP or BCE was added, obtaining BCP-S and BCE-S respectively. For the fat model, 4 g of olive oil were mixed with water (up to 100 mL) during 1 min using a homogenizer (T18 digital ULTRA-TURRAX, IKA, Staufen, Germany) at 13,000 rpm, and BCP or BCE was added, obtaining BCP-O and BCE-O, respectively. For the protein model, 4 g of whey protein were dissolved in distilled water (up to 100 mL) and BCP or BCE was added, obtaining BCP-WP and BCE-WP, respectively. Finally, for the fifth model system, 5.5 g of starch was gelatinized in 70 mL of distilled water, as described before. When it was cooled, 4 g of oil and 4 g of whey protein were added in distilled water up to 100 mL. Lastly, the solution was mixed for 1 min in the homogenizer at 13,000 rpm and BCP or BCE was added, obtaining BCP-All and BCE-All, respectively.

### 2.4. Microstructure

The microstructure analysis was conducted following Hernández-Carrión et al. [37], with some modifications. A Nikon ECLIPSE 80i light microscope (Nikon Co., Ltd., Tokyo, Japan) was used working in the bright field and fluorescence modes. The autofluorescence of the phenolic compounds in the samples was observed using a mercury arc lamp with a tetramethyl rhodamine filter (λ_ex_ = 543/22 nm, λ_em_ = 593/40 nm) as the excitation source. Samples were visualized using 10× and 20× objective lenses. The images were captured and stored at 1280 × 1024 pixels using the microscope software (NIS-Elements F, Version 4.0, Nikon, Tokyo, Japan).

### 2.5. Analytical Determinations

To determine the content of phenolic compounds and its AC, the model systems were centrifuged for 20 min at 5000× *g* and 4 °C and then filtered (Whatman^®^ Grade 4, Fisher Scientific S.L., Madrid, Spain). The supernatant was used for the analysis.

#### 2.5.1. Total Phenolic Content (TPC)

TPC was determined by the Folin–Ciocalteu (F-C) assay according to the procedure described by Singleton et al. [38], with some modifications. In test tubes, 6 mL of double-distilled water and 1 mL of the supernatant were added (1 mL of double-distilled water for the blank). Then, 0.5 mL of Folin–Ciocalteu reagent (1:1 (*v*/*v*)) was added. After 3 min, 1 mL of sodium carbonate solution (20% (*w*/*v*)) and 1.5 mL of distilled water were added and vortexed. The mixture was kept at room temperature (≈25 °C) in a dark room for 90 min. Absorbance was measured at 765 nm. The absorbances obtained were related to a calibration curve, which was prepared using different concentrations of gallic acid. The results were expressed as mg of Gallic Acid Equivalents (GAE) per 100 mL of sample. The analysis was made in triplicate.

#### 2.5.2. Antioxidant Capacity (AC)

The AC was measured by the ferric reducing antioxidant power assay (FRAP) described by Benzie and Stain [39] and Pulido et al. [40] with slight modifications. Following the order described below, 30 μL distilled water, 30 μL extract, and 900 μL FRAP reagent were added in 1.5 mL cuvettes. Distilled water was used as blank. The cuvettes were incubated in a bath at 37 °C for 30 min and the absorbance was measured at 595 nm. The absorbances obtained were related to a calibration curve, which was prepared using different concentrations of Trolox. The results were expressed as µmol Trolox per mL of sample. The analysis was made in triplicate.

### 2.6. Simulated In Vitro Digestion Process

An in vitro gastrointestinal tract model was used to simulate the biological fate of the ingested samples following the methodology described by Minekus et al. [41], with modifications [42,43]. Three phases were simulated: oral, gastric, and intestinal.

The digestion process was conducted in a “Carousel 6 Plus” reaction station (Radleys, United Kingdom). To mimic human physiological conditions, the analysis was conducted with a controlled temperature (37 °C) and agitation (150 rpm), and without light. Both the gastric and intestinal step were performed in a N_2_ atmosphere to mimic human physiological reduction of oxygen levels during digestion [42].

Solutions of simulated salivary fluid (SSF), simulated gastric fluid (SGF), and simulated intestinal fluid (SIF) were prepared according to the compositions described by Minekus et al. [41]. First, for the oral stage, 5 mL of the sample were added in the digestion flask. Then, 4 mL of SSF + α-amylase, 19 μL of CaCl_2_, and 0.981 mL of distilled water were added. The pH of the saliva was adjusted to 7. The oral digesta was under agitation for 2 min at 37 °C. Second, for the gastric phase, 8 mL of SGF + pepsin and 4 μL of CaCl_2_ were added. The pH was adjusted to 3 using 1M HCl, and the volume of distilled water necessary for a total volume of 10 mL was added. The mixture was incubated at 37 °C for 1 h under agitation without oxygen. Third, for the intestinal stage, 5.3 mL of SIF + pancreatin, 40 μL of CaCl_2_, 5.3 mL of SIF + bile salts [43], and 5.3 mL of SIF + lipase were added. The pH was adjusted to 7 using 1M HCl or 1M NaOH. Once the pH was adjusted, the volume of distilled water necessary for a total volume of 30 mL was added; the pH was readjusted to 7. Finally, for the filtration process, the final digestion mixture was centrifuged (27,641× *g*, 20 min, 4 °C) and filtered (Whatman^®^ Grade 4). The residue was the non-digested fraction, and the filtered solution was the soluble fraction available for absorption. The samples were stored at −80 °C until further analysis. The digestions were conducted twice.

### 2.7. Bioaccessibility

The bioaccessibility (BA) is defined as the amount of an ingested nutrient available for absorption in the gut after digestion and is calculated using Equation (1) [18,44,45].
Bioaccessibility (%) = (TPC of the soluble fraction / TPC of fresh samples) × 100(1)

### 2.8. Statistical Analysis

A categorical multifactorial experimental design with two factors, namely, the source of polyphenol (pomace, BCP or extract, BCE) and model system, was performed on the values of TPC and AC. Analysis of variance (ANOVA) was performed on the data. Least significant difference (LSD) Fisher’s tests were used to evaluate the mean value differences (*p* < 0.05) using XLSTAT 2014 statistical software (Microsoft, Mountain View, CA, USA).

## 3. Results and Discussion

### 3.1. Microstructure of the Model Systems

Figure 1 shows the light microscopy (LM) and fluorescence (FL) images of the fresh (non-digested) systems.

The BCP-Wa model system (Figure 1A) comprises pomace particles of different sizes and shapes distributed in a continuous phase of water. These particles are mostly reddish and brownish. Blackcurrant pomace is rich in polyphenols, mainly anthocyanins [9], and the aglycone form of anthocyanins are auto fluorescent [46,47]. The intensity of the intrinsic fluorescence of the pomace is observed in Figure 1B. In the BCE-Wa system (Figure 1C), the extract was dissolved in the continuous phase and coloration cannot be observed, nor was any important autofluorescence observed (Figure 1D).

In the BCP-S systems (Figure 1E), the pomace particles are distributed in the continuous phase formed by partially gelatinized starch granules. In these systems, pomace particles maintain their characteristic reddish coloration (Figure 1E) and autofluorescence (Figure 1F). In BCE-S, where the extract was added to the starch system, some colored gelatinized granules can be observed (Figure 1G), and a slight fluorescence (Figure 1H) is also detected. This could be indicating interactions between the starch polymers and polyphenols. 

In BCP-O systems (Figure 1I), the pomace particles maintain their reddish coloration and some oil globules can be observed, containing small pomace particles inside. Moreover, in these systems, both particles and globules show autofluorescence (Figure 1J). However, this cannot be observed in BCE-O systems (Figure 1K,L). Thus, while pomace is interacting with the oil, the extract is diluted into the liquid media, and interactions with oil are not produced.

When the pomace is incorporated into a system with protein (BCP-WP and BCP-All, Figure 1M,Q, respectively), discolored pomace particles distributed in the continuous phase can be observed. This can be due to the lixiviation of phenolic compounds from the pomace to the continuous phase. In these systems, pomace particles present low levels of autofluorescence (Figure 1N,R) if compared with systems prepared with starch and oil. However, neither color nor fluorescence are observed in the BCE model systems (Figure 1O,P,S,T). Strong protein–polyphenol interactions could be favoring the extraction of the polyphenols from the pomace toward the medium, becoming part of the continuous phase. However, the polyphenols in the extract could be dissolved into the aqueous medium, preventing color or autofluorescence from being observed.

### 3.2. Total Phenolic Content (TPC) and Antioxidant Capacity (AC) of the Model Systems: Analysis of Macronutrient–Polyphenol Interactions

A two-way ANOVA was performed for the TPC and AC results. In both analyses, the results show significant interactions (*p* < 0.05) between the two factors: source of polyphenol (BCP or BCE) and model system (Wa, S, O, WP, and All). In addition, each factor presented a significant effect. The results can be seen in Figure 2. 

When comparing the use of the pomace (BCP) or the extract (BCE) for TPC results (Figure 2A), samples prepared with BCP gave significantly (*p* < 0.05) lower values than those prepared with BCE for the model systems prepared with water (BCP-Wa and BCE-Wa) or oil (BCP-O and BCE-O). Regarding the different model systems, the formulations with the highest values for both BCP and BCE were those prepared with protein (BCP-WP and BCE-WP, and BCP-All and BCE-All). For the starch models, BCP-S did not present significant differences (*p* > 0.05) with the water model (BCP-Wa) and the oil model (BCP-O), but BCE-S did (*p* < 0.05).

As it can be observed, for the AC results (Figure 2B), that formulations with O (BCP-O and BCE-O) and formulations with Wa (BCP-Wa and BCE-Wa) presented the same trend as the TPC results. In addition, formulations with WP in their composition (BCP-WP and BCE-WP, and BCP-All and BCE-All) also presented the same trend but the difference between the results for these formulations and the remaining model systems decreased when compared with TPC. The lowest results for AC corresponded to BCP-O and the highest to BCE-Wa.

The different TPC and AC results for BCP-Wa and BCE-Wa (Figure 2) can be explained because, in the BCP, the polyphenols are associated with the pomace fiber compounds through covalent interactions or weak non-covalent interactions (hydrogen bonds, Van der Waal forces, and hydrophobic interactions), avoiding its extraction for its determination [11,48]. In contrast, for BCE, its behavior is different. Phenolic compounds from BCE are dissolved into the water because besides the distinctive aromatic rings, these compounds have a substantial number of hydroxyl groups that gives them a highly polar structure, allowing its solubilization in water. Thus, for BCP, the phenolic compounds are bound to the pomace whereas the phenolic compounds in BCE are free to be dissolved, as shown in the schematic representation of Figure 3. In Figure 1A, the red-colored pomace particles can be observed, whereas in Figure 1C there are discolored particles. It confirms that polyphenols from pomace are not readily available for interaction and that polyphenols in the extract are soluble into the media and free to interact with other compounds or macromolecules.

Regarding the TPC and AC results for starch formulations, BCP-S and BCP-Wa models show no differences between them. Thus, incorporating starch in BCP systems did not cause critical changes. However, BCE-S presented lower results (*p* < 0.05) than the BCE-Wa. It could mean there are interactions between the starch and the free polyphenols from BCE because of their high availability to interact. Colored starch granules can be observed in Figure 1G, which could confirm the existence of these interactions. Some authors, such as Amoako and Akiwa [22] and Zhu et al. [23], have described the interactions between starch and polyphenols. Two types of interactions between amylose and polyphenols are described: a V-type amylose inclusion complex, which is driven by hydrophobic interactions, and a non-inclusion complex, driven by the hydrogen bounds between the polyphenol hydroxyl groups and the hydrophilic part of the amylose chains [23,49,50,51]. A schematic representation of the amylose–polyphenol interactions is shown in Figure 3. The inclusion complex may not be possible with the bulky-sized polyphenols because of the limitation of the size of the cavity in the amylose helix, but possible with the low-molecular-weight polyphenols [23,50]. Thus, as the extract comprises polyphenols of different chemical structure and size, both types of interactions may occur simultaneously: inclusion-type interactions for small-sized polyphenols and the non-inclusion type for bigger polyphenol molecules.

When comparing the BCP and BCE values of the oil systems (BCP-O and BCE-O), the results are higher when oil is formulated with BCE (Figure 2), probably because of the hydrophilic nature of the free polyphenols that causes their dissolution in the aqueous medium, avoiding its interaction with oil. In addition, an effect of micellarization of phenolic compounds with oil can be produced in the BCP-O systems, i.e., phenolic compounds bounded to pomace fiber are within a micelle covered by oil, as described by Ortega et al. [24]. Some components of pomace, as seeds and peels, have a hydrophobic nature that allows the micellarization. This effect is showed in the images of Figure 1E, in which encapsulated pomace fractions into the oil globules can be observed (schematically represented in Figure 3). Therefore, if polyphenols are participating in the micelles, the phenolic content and the antioxidant capacity are reduced.

The formulations with the highest TPC and AC values for both BCP and BCE were those prepared with protein (BCP-WP and BCE-WP, and BCP-All and BCE-All) (Figure 2). The interaction between polyphenols and milk proteins can be caused by both non-covalent and covalent forces [14,15,26,27,52]. Non-covalent interactions involve weak associations, specifically a combination of hydrogen bonds and hydrophobic interactions. The latter are reversible interactions that would involve aromatics rings of polyphenols and hydrophobic sites of proteins, whereas hydrogen bonding occurs between hydrogen-acceptor sites of the proteins and the hydroxyl groups of the polyphenols. These interactions will be determined by the molecular weight, conformational mobility, and flexibility, as well as by the relation between the donor/acceptor hydrogen bonds in the proteins and polyphenols [14,15]. In addition, protein and polyphenols can bind through covalent bonds. These bonds are formed between O-quinones, coming from the oxidation of BCP or BCE polyphenols and nucleophilic groups of proteins, such as -NH_2_ and -SH; these interactions are irreversible. Most common interactions are the non-covalent types, such as hydrogen bonding and hydrophobic forces, which can lead to a complex formation between WP and polyphenols [53,54,55]. In addition, the great molecular weight and high polarity of most polyphenols from BCP and BCE favors the interactions because of the high number of binding sites [26,52]. Both types of interactions—non-covalent and covalent—are represented in Figure 3.

In the BCP-WP and BCP-All model systems (Figure 1M,Q), the particles from the BCP are discolored, probably because of the release of the polyphenols from the pomace matrix to the medium. The non-covalent hydrophobic interactions between the aromatic ring of polyphenols and the hydrophobic sites of the proteins, which leads to the complexation with whey proteins, could force the release of polyphenols from the pomace. This effect is also observed in Figure 1N,R because there is a low fluorescence intensity. This is because the fluorescence phenomenon is given by the aromatic rings, and if there is a complex formation where the aromatic ring is involved, the fluorescence intensity is lower. However, despite this complex formation, the TPC values obtained for all the systems prepared with protein are high (Figure 2A) because the hydroxyl groups still possess their reducing capacity. In addition, it is also necessary to consider that the high TPC results could be related to the detection of the aromatic amino acids with reducing capacity present in the whey protein, detected by the Folin–Ciocalteu reagent [56,57,58].

Regarding formulations with protein and BCE (BCE-WP and BCE-All), a complex formation by non-covalent interactions through hydrogen bonding is also likely. These interactions are because of the higher availability of the free phenolic extract compounds to interact, which facilitates the interaction with the globular WP.

### 3.3. Total Phenolic Content, Antioxidant Capacity, and Bioaccessibility of the Digested Model Systems

Figure 4 shows there are significant interactions (*p* < 0.05) between the different model systems and using pomace or extract for both the TPC and AC analysis. In addition, each factor has a significant effect.

In terms of the differences between using BCP or BCE as a polyphenol source, for TPC (Figure 3), no significant differences (*p* > 0.05) were observed between the water models (BCP-Wa or BCE-Wa) and the models formulated with starch or oil (BCP-S, BCE-S, BCP-O, and BCP-S). However, when WP was used as an ingredient, the TPC was lower (*p* < 0.05) when the system was formulated with BCE, whereas when all the ingredients were added to the model, the TPC was higher (*p* < 0.05). Finally, results for models with protein in their formulation (BCP-WP, BCE-WP, BCP-All, and BCE-All) were the highest.

For AC values (Figure 4B), and regarding the differences between using BCP or BCE, the formulations with all the ingredients (BCP-All and BCE-All) presented significant (*p* < 0.05) differences, with values for BCP-All higher than from BCE-All. The rest of the formulations did not show significant differences (*p* > 0.05) whether BCP or BCE were used. In addition, formulations with BCP did not show differences (*p* > 0.05) between the BCP-Wa model and models BCP-S and BCP-O. Furthermore, a similar trend than for TPC results can be observed, with the models formulated with protein being the ones with the highest results. However, for the BCE results in systems elaborated with protein (BCE-WP and BCE-All), the results are the highest (*p* < 0.05).

The bioaccessibility results (Figure 5) allow to understand the changes that occurred in the interactions between the macronutrients and polyphenols induced by the conditions throughout the in vitro digestion process. The release, stability, and solubility of the polyphenols is related to the food matrix composition and the polyphenols’ chemical structure [59]. The bioaccessibility of each class of polyphenol is affected by the molecular weight, glycosylation (aglycones are more hydrophilic and thus more easily absorbed), the interactions between the polyphenols and food components, and also by its different pH transformations [19,60].

Figure 5 shows that the samples formulated with BCP had the highest bioaccessibility results, which exceeded the 100% level for the BCP-Wa, BCP-S, and BCP-WP samples, or were near to 100% for BCP-O.

Thus, bioaccessibility results above 100% could mean there is a release of polyphenols from the pomace during the in vitro digestion process, allowing them to be available for further absorption into the systemic circulation. Moreover, during the digestion process, the phenolic compounds are released from the food matrix in the upper part of the gastrointestinal tract because of the polyphenol solubilization into the intestinal fluids at physiological conditions, where they become available for intestinal absorption [11]. At the end of the gastrointestinal tract, there would be two fractions of phenolic compounds: the accessible phenolic compounds with a low molecular weight, which may be partially absorbed through the small intestine mucosa, and the non-accessible phenolic compounds, which comprises high-molecular-weight polyphenols or low-molecular-weight-polyphenols bonded to dietary fiber and/or trapped into the fiber matrix core [11,29,48]. 

However, the models with BCE, especially the one formulated only with water (BCE-Wa), had the lowest results compared with the rest of models. Several authors [17,33,34] have described a decrease in bioaccessibility of polyphenols, particularly in anthocyanins, after the digestion process. This may be due because the polyphenols in the extract are free and dissolved into the medium once the simulated digestion starts, being more susceptible to changes and therefore to their degradation. Specifically, under the conditions during the simulated digestion (mainly changes in pH and enzyme action), the red flavylium cation is transformed into less stable forms, such as the pseudobase, quinoidal base, and chalcone [17].

Despite BCP-O not having a BA over 100%, its result is 94%; so, although there is not a significant release of polyphenols during in vitro digestion, important polyphenol degradation is not occurring either. Gu et al. [2] proposed a mechanism of anthocyanin protection, which is one of the main phenolic compounds of blackcurrant pomace, based on the anthocyanin incorporation into the lipid phase of the micelles (bile salts from digestion emulsify lipids and break into micelles under the actions of lipase before absorption). Thus, the oil may exert a protective effect on polyphenols.

The highest BA results were for BCP-WP. As described in Section 3.2, a protein–polyphenol complex is formed through hydrophobic interactions. This complexation could have a protective role over the polyphenols, avoiding their transformation by alkaline pH or their oxidation into O-quinones by reactive oxygen species [54,61]. In addition, during the digestion process, a release of polyphenols from the pomace because of gastrointestinal conditions is produced. Therefore, new complexes may form, and WP would exert a protective effect on the polyphenols from blackcurrants during in vitro digestion. As described by Meng et al. [55] and Tagliazucchi et al. [61], milk proteins could avoid autoxidative reactions through the gastrointestinal tract, acting as a carrier for the delivery of antioxidant compounds. 

Finally, BCP-All presented the lowest values compared with other models with BCP. Here, all the ingredients are used in the formulation, giving the possibility of multiple interactions between the ingredients and the phenolic compounds. This effect could form different complexes or aggregates of high molecular weight, which would avoid their absorption through the epithelium [34]. Thus, polyphenols embedded into the complex could reach the colon unaltered, where the phenolic compound may be absorbed intact or metabolized by colonic microbiota. Furthermore, polyphenols can stimulate beneficial bacteria and inhibit pathogenic bacteria, thus resulting in great importance for colon health [62].

## 4. Conclusions

When polyphenols are incorporated into the model systems as blackcurrant pomace, their bioaccessibility is increased if compared to the polyphenols added from the extract source. The bounding of polyphenols to pomace and the interactions that take place with the nutrients of the model systems influence the proportion of polyphenols that can be available for absorption in the initial stages of intestinal digestion. However, the polyphenols in the extract are dissolved into the aqueous medium, which makes them more susceptible to changes, decreasing their bioaccessibility. When the model systems are formulated with pomace and only one nutrient, the bioaccessibility is higher than when the system contains all the nutrients. Of all the components studied, the greatest effect on bioaccessibility is observed for proteins due to the protein–polyphenol complex formation. Although the data obtained with these model systems cannot be directly extrapolated to human in vivo conditions, they could be a helpful approach for determining the effects of the food matrix on polyphenol bioaccessibility. Further in vivo investigations would be helpful to better comprehend the fate of polyphenols in the last steps of the digestion process.

## Figures and Tables

**Figure 1 foods-10-00847-f001:**
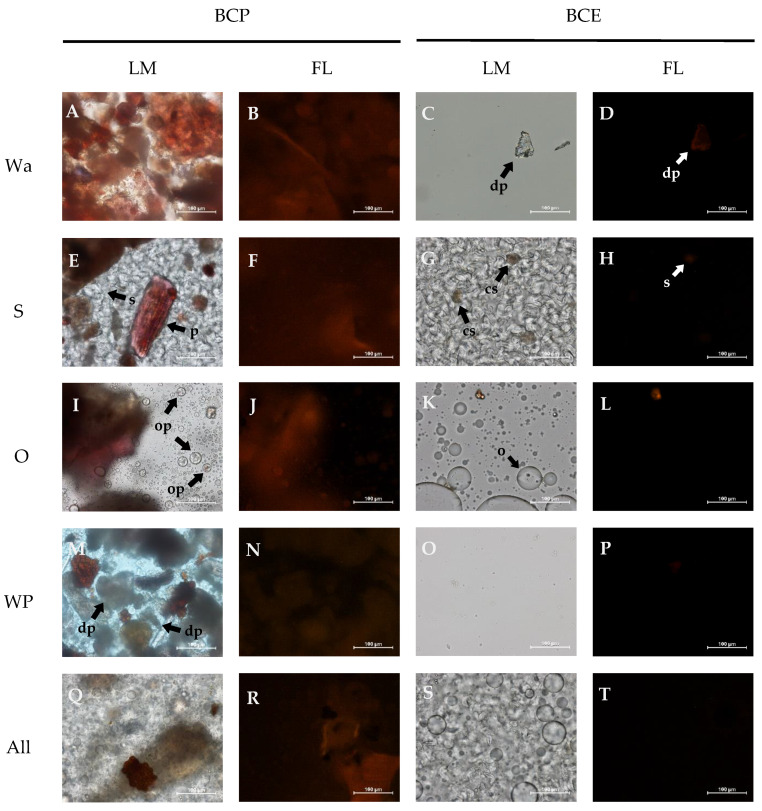
Light microscopy (LM) and fluorescence (FL) pictures of the model systems at 100 µm. BCP: blackcurrant pomace; BCE: blackcurrant extract; Wa: water; S: starch; O: oil; WP: whey protein; and All: water, starch, oil, and whey protein. BCP-Wa (**A**,**B**), BCE-Wa (**C**,**D**), BCP-S (**E**,**F**), BCE-S (**G**,**H**), BCP-O (**I**,**J**), BCE-O (**K**,**L**), BCP-WP (**M**,**N**), BCE-WP (**O**,**P**), BCP-All (**Q**,**R**) and BCE-All (**S**,**T**), where first letter between parenthesis correspond to LM images and the second to FL images. Arrows: p (pomace), s (starch granules), cs (colored starch granules), op (oil globules with pomace), o (oil droplets), and dp (discolored pomace).

**Figure 2 foods-10-00847-f002:**
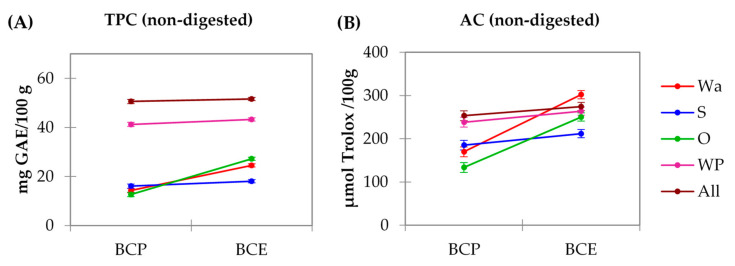
Interaction plot between the source of polyphenol and model system for (**A**) total phenolic content (TPC) and (**B**) antioxidant capacity (AC). BCP: blackcurrant pomace; BCE: blackcurrant extract; Wa: water; S: starch; O: oil; WP: whey protein; and All: water, starch, oil, and whey protein.

**Figure 3 foods-10-00847-f003:**
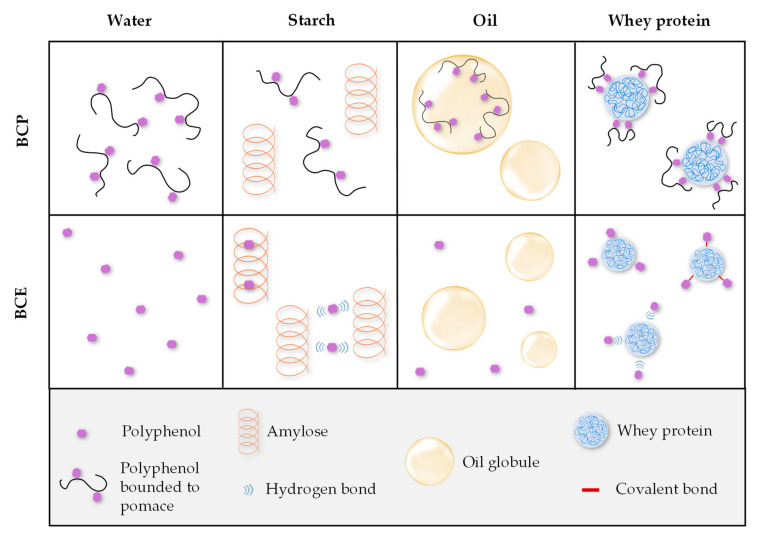
Schematic representation of the polyphenol–macronutrient interactions. BCP: blackcurrant pomace; BCE: blackcurrant extract.

**Figure 4 foods-10-00847-f004:**
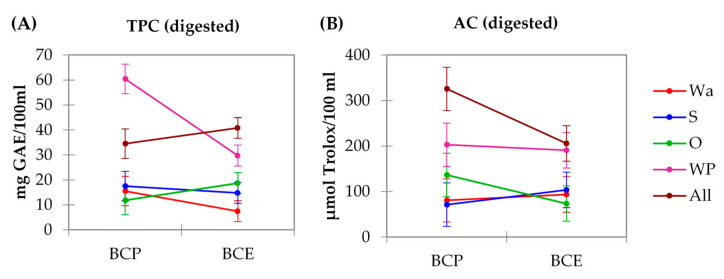
Interaction plot between the source of polyphenol and model system for (**A**) total phenolic content (TPC) and (**B**) antioxidant capacity (AC). BCP: blackcurrant pomace; BCE: blackcurrant extract; Wa: water; S: starch; O: oil; WP: whey protein; and All: water, starch, oil, and whey protein.

**Figure 5 foods-10-00847-f005:**
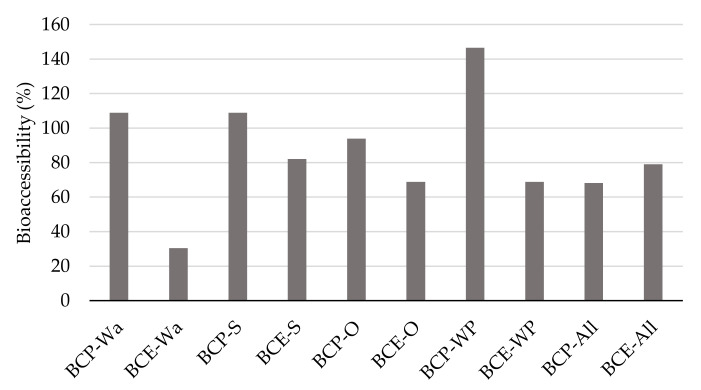
Bioaccessibility results from all the model systems. BCP: blackcurrant pomace; BCE: blackcurrant extract; Wa: water; S: starch; O: oil; WP: whey protein; and All: water, starch, oil, and whey protein.

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
