# Peer review of "Interactions between Blackcurrant Polyphenols and Food Macronutrients in Model Systems: In Vitro Digestion Studies"

_foods, 2021, doi:10.3390/foods10040847_

Round 1

Reviewer 1 Report

In this study Diez-Sanchez et al. report the influence of food macronutrients on bioaccessibility of polyphenols from blackcurrant pomace, a byproduct of  industrial food processing. Interactions between polyphenols and food macronutrients were examined through in vitro digestion models. Variations in polyphenols bioaccessibility were detected when polyphenols were added as pomace or as polyphenolic pomace extract to the digestion model system. Furthermore, bioaccessibility was found increased when digestion was performed with pomace and only one nutrient as compared to more complex food matrices with proteins showing the greatest effect on polyphenols bioaccessibility with respect to other macronutrients.

The manuscript is generally well written and clearly presented and the experimental design is appropriate. 

However there are some minor points of weakness that should be addressed before considering the manuscript suitable for publication, with particular regard to technical details that should be expanded and clarified as follows:

1) Material and Methods section: authors should provide more details regarding sample preparation before extraction: have the samples been lyophilized or ground?

2) paragraph 2.5.2: the antioxidant activity was only detected with the FRAP assay. Authors should explain why other methods (ABTS and/or DPPH) were excluded from this study.

3) Language editing is required to fix grammar errors (see, for example, lanes 157, 184, 228)

Author Response

1) Material and Methods section: authors should provide more details regarding sample preparation before extraction: have the samples been lyophilized or ground?

According to the reviewer suggestion, the information has been added in L102-105 and L114.

2) paragraph 2.5.2: the antioxidant activity was only detected with the FRAP assay. Authors should explain why other methods (ABTS and/or DPPH) were excluded from this study.

We made some tests with DPPH methodology, but the results were not concluding due to the type of matrix. The samples caused interferences that prevented the analysis from being carried out correctly, specifically in digested samples. In addition, we wanted to focus our results on the interactions and the changes in the phenolic content, therefore we decided not to use another method but FRAP for measuring the antioxidant capacity.

3) Language editing is required to fix grammar errors (see, for example, lanes 157, 184, 228)

The first manuscript was revised by a native English-speaking editor. Even though, according to the reviewer suggestion, the R1 manuscript has been revised again. The recommendations have been amended in L159, L186 and L230. We attach the editing certificate.

Reviewer 2 Report

The publication is interesting and shows an important problem of compounds _ polyphenols can reduce the risk of many diseases. Because the health benefits of polyphenols are based on the compounds' bioavailability. The evidence collected in the present review suggests that when plant phenols are consumed along with food macronutrients, the bioavailability and bioactivity of polyphenols can be significantly affected. 

Author Response

The publication is interesting and shows an important problem of compounds _ polyphenols can reduce the risk of many diseases. Because the health benefits of polyphenols are based on the compounds' bioavailability. The evidence collected in the present review suggests that when plant phenols are consumed along with food macronutrients, the bioavailability and bioactivity of polyphenols can be significantly affected.

The authors would like to thank the reviewer for the comment. As said by the reviewer, we tried to show that the interactions between polyphenols and macronutrients need to be taken into account when the bioaccessibility of polyphenols is analyzed.

Reviewer 3 Report

Dear Editor and Authors

Congratulations on the study performed in

Interactions Between Blackcurrant Polyphenols and Food Macronutrients in Model Systems. In Vitro Digestion Studies.

The work is very well organized and written. The data provides relevant information for the Industry which utilizes this kind of product.

Minor improvements:

In the introduction, in my point of the view, the authors should include the name of the main anthocyanins in this matrix. They even cited References that contain this information.

In the discussion, some details should be added to provide input about the possible interference of the acidic extraction of the anthocyanins, especially if they are in the aglycon form, resulting from the production of the pomade. In fact, some of this correlated data is discussed in the gastric phase.

Line 295-296 the name of the authors should be changed for the number ones [ ]

Line 447 in vivo should be changed for in vivo

My best regards,

Author Response

In the introduction, in my point of the view, the authors should include the name of the main anthocyanins in this matrix. They even cited References that contain this information.

According to the reviewer suggestion, the main anthocyanins of blackcurrant pomace have been added in L35 and L36.

In the discussion, some details should be added to provide input about the possible interference of the acidic extraction of the anthocyanins, especially if they are in the aglycon form, resulting from the production of the pomade. In fact, some of this correlated data is discussed in the gastric phase.

The extraction carried out for extract obtention was not made in acidic conditions, only with ethanol 60% at 60 ⁰C. Additionally, the main anthocyanins in the blackcurrant pomace are present as glycosides (this information has been added in the introduction section of the manuscript as previously required by the reviewer). So, we prefer not including the idea about the possible interference of the acidic conditions.

Line 295-296 the name of the authors should be changed for the number ones [ ]

It was a mistake; the reference style has been changed in L298.

Line 447 in vivo should be changed for in vivo

According the reviewer suggestion, the text has been changed in L450.